# Hydrothermal Synthesis of WO_3_·0.33H_2_O Nanorod Bundles as a Highly Sensitive Cyclohexene Sensor

**DOI:** 10.3390/s19051257

**Published:** 2019-03-12

**Authors:** Xiaofei Liu, Xintai Su, Chao Yang, Kongjun Ma

**Affiliations:** 1Ministry Key Laboratory of Oil and Gas Fine Chemicals, College of Chemistry and Chemical Engineering, Xinjiang University, Urumqi 830046, China; lxf40414@163.com (X.L.); jarryyang1924@163.com (C.Y.); 2The Key Laboratory of Pollution Control and Ecosystem Restoration in Industry Clusters (Ministry of Education), School of Environment and Energy, South China University of Technology, Guangzhou 510006, China; suxintai827@163.com

**Keywords:** hydrothermal, WO_3_·0.33H_2_O, p-aminobenzoic acid, nanorod bundles, cyclohexene, gas sensing

## Abstract

In this paper, WO_3_·0.33H_2_O nanorods were prepared through a simple hydrothermal method using p-aminobenzoic acid (PABA) as an auxiliary reagent. X-ray diffraction (XRD) and transmission electron microscopy (TEM) images showed that the products with PABA addition were orthorhombic WO_3_·0.33H_2_O, which were mainly composed of nanorods with different crystal planes. The sensing performance of WO_3_·0.33H_2_O nanorod bundles prepared by the addition of PABA (100 ppm cyclohexene, Ra/Rg = 50.6) was found to be better than the WO_3_ synthesized without PABA (100 ppm cyclohexene, Ra/Rg = 1.3) for the detection of cyclohexene. The new synthesis route and sensing characteristics of as-synthesized WO_3_·0.33H_2_O nanorods revealed a promising candidate for the preparation of the cost-effective gas sensors.

## 1. Introduction

With the development of modern industry and society, people pay more and more attention to safety and the environment. The detection and warning of volatile compounds has become an important concern [1,2]. Cyclohexene is a toxic and flammable compound, which is commonly used in the production of fine chemical products such as adipic acid, nylon-6, and nylon-66, and is widely used as an important material in the organic synthesis of other chemicals [3,4]. However, cyclohexene vapor stimulates the eyes, skin, and respiratory tract, which may affect the central nervous system. Currently, the gas-sensitive materials that can be used for the determination of cyclohexene vapor are still very limited; therefore, gas sensors for detecting cyclohexene vapor is very important.

In recent years, tungsten oxide-based materials have been proven to be an effective sensitive material for the detection of both reducing and oxidizing gases [5,6]. WO_3_·xH_2_O (x = 0, 0.33, 0.5, 1, 2), an important tungsten oxide-based material, has been found to exhibit a good response to cyclohexene [7]. So far, WO_3_·xH_2_O nanomaterials have been successfully synthesized using a hydrothermal method [8,9,10,11,12,13,14,15]. Soluble organic acids, such as citric acid, tartaric acid, and ascorbic acid, are widely used in the hydrothermal synthesis of tungsten oxide [16,17,18]. These organic acids have important effects on the phase morphology and size of tungsten oxide. However, some poorly soluble organic acids, such as p-aminobenzoic acid (PABA), have been reported to be used as weak reductants and structural directing agents for the hydrothermal synthesis of W_18_O_49_ nanowire networks [19]. Considering this information, we here investigate what can be achieved by adding PABA to our experiments of the synthesis of WO_3_·0.33H_2_O.

In this paper, WO_3_·0.33H_2_O nanorods were synthesized by a hydrothermal method using PABA as a regulator. When only hydrochloric acid was used to control the pH value and PABA was not used, the product was short WO_3_ nanorods. When different amounts of PABA were used as auxiliary reagents, WO_3_·0.33H_2_O nanorods with different crystal faces were obtained. Among them, the product obtained with 8 g of PABA had the best gas-sensitive response to cyclohexene gas. This method of the hydrothermal synthesis of tungsten oxide using poorly soluble organic matter provided the potential to obtain high-performance gas-sensing materials for the detection of cyclohexene organics.

## 2. Experimental

### 2.1. Preparation of WO_3_·0.33H_2_O Nanorods

Different amounts of PABA were used to prepare WO_3_·0.33H_2_O nanorods via a hydrothermal process. First, 1 g of Na_2_WO_4_·2H_2_O was dissolved in 15 mL of water, and a certain amount of PABA (0/4/8/12 g) and NaOH (0/1.17/2.33/3.51 g), at a 1:1 molar ratio, was mixed into 15 mL of deionized water to form a sodium p-aminobenzoate solution. Second, the above two solutions were mixed together under stirring, and the pH of the solution was adjusted to 1 with 6 M HCl solution. At this pH, tungstic acid precipitation is fully formed. Then, water was added to obtain a total volume of 60 mL. Finally, the white mixture was put into a 100 mL Teflon-line autoclave before being sealed and heated at 180 °C for 15 h. After cooling the autoclave to room temperature, the solid product was centrifuged and washed three times with water and ethanol, and dried at room temperature for 24 h. The samples prepared with different amounts of PABA (0, 4, 8, and 16 g) were noted as S1, S2, S3, and S4.

### 2.2. Characterization

The X-ray diffraction of the as-prepared sample was characterized by a DX-1000 X-ray diffractometer (DX-1000, Dandong Fangyuan Instrument Co. Ltd., Dandong City, Liaoning Province, China), equipped at a scanning rate of 2° min^−1^ and ranging from 10° to 80° with graphite-monochromatized Cu-K ray radiation (=1.5418 Å). Transmission electron microscope (TEM) images of the product were obtained on a Hitachi H-600 transmission electron microscope with an accelerating voltage of 100 kV. The field-emission scanning electron microscopy (FESEM) images were obtained on a Hitachi S4800. 

### 2.3. Fabrication of Gas Sensors

Gas-sensing performance was evaluated in a commercial gas-sensing measurement system WS-30A (Zhengzhou, China). The method and instruments of WO_3_·0.33H_2_O-based gas sensors were similar to our previous report [20]. The WO_3_·0.33H_2_O sample was dispersed in ethanol to form a suspension and then coated on alumina ceramic tubes with a pair of Au electrodes. The structural sketch of the WO_3_·0.33H_2_O-based gas sensor is shown in Figure 1a. There was no additional binder, except for ethanol. Finally, the device was placed in the gas test system and heated for 3 days by a resistance heater at 300 °C to improve the response stability of the detected gas. A stationary state gas distribution method was used to test the gas-sensing properties [21]. The gas-sensing test was conducted with a fixed relative humidity of 30% and the balance gas was air in the testing chamber. The measuring electric circuit of the gas sensor is shown in Figure 1b.

Taking ethanol as an example, a certain amount of ethanol was injected into the test chamber by a micro-injector. The ethanol vapor concentrations (10–1000 ppm) were calculated based on the concentration of the ethanol and the volume of the chamber. The amounts of liquid ethanol (*V*_ethanol, L) were determined according to Equation (1).
(1)$$V_{ethanol}=\frac{10^{-9}V_{0}MC_{ethanol}}{22.4\rho p}$$

Here, $V_{0}$ is the volume of the chamber ($V_{0}=\;18\;L$), M is the mole mass ($g\;\cdot {\mathrm{mol}}^{-1}$) of ethanol, $C_{ethanol}$ is the concentration of ethanol (ppm), ρ is the density of ethanol ($g\cdot {\;\mathrm{cm}}^{-3}$), and p is the rate of purity of ethanol. The operating temperature ranges from 260 °C to 370 °C, controlled by an electric heating system. The WO_3_·0.33H_2_O sensor (R) is connected to a load resistor ($R_{L}$) with a known resistance (1 MΩ), and a source voltage ($V_{c}$) of 5 V is loaded on the circuit. The system measured the voltages ($V_{out}$) loaded on the resistor $R_{L}$, and the resistances (R) of the WO_3_·0.33H_2_O sensor were calculated according to Equation (2). Here, the response ($R_{r}$) is defined as Equation (3), where $R_{a}$ and $R_{g}$ are the resistances of the WO_3_·0.33H_2_O sensor in air and in ambient ethanol, respectively.
(2)$$R=\frac{V_{c}-V_{out}}{V_{out}}\times R_{L}$$
(3)$$R_{r}=\frac{R_{a}}{R_{g}}$$

## 3. Results and Discussion

### 3.1. Structure Characterization of Gas-Sensing Materials

The typical TEM images of the four samples are shown in Figure 2. There are a lot of irregular rectangular nanorods and a few nanorods, about 10 nm, seen in the S1 sample, which is prepared without PABA as an auxiliary agent (Figure 2a). When 4 g or 8 g of PABA is used as the auxiliary reagent in the hydrothermal system, the morphology of the products changes, resulting in a large rod structure (Figure 2b,c). These nanorods can reach hundreds of nanometers in length and 10–50 nanometers in diameter. As can be seen from the diagram, some of the nanorods are stacked in parallel in the center, and some of the nanorods are assembled in bunches. This result shows the strong surface interaction capability of such nanorods. Figure 3a–c show the element distribution of S3, from which we can see that S3 contains two elements of W and O, and W and O are evenly distributed on the nanorods. Figure 3d is the SEM image of S3 with ethanol as a binder. We can observe that, with the S3 load on the gas sensor, the nanoparticles are stacked together to form the nanorod bundles. When the additional amount of PABA reaches 12 g, the morphology of the product is large aggregates and irregular particles (Figure 2d). More interestingly, these four samples show different lattices. The HRTEM image and corresponding fast Fourier transform (FFT) pattern (Figure 2f–h) demonstrate that the spacings between adjacent lattice planes of the four samples are 0.383 nm, 0.376 nm, 0.367 nm, and 0.490 nm, respectively. These exposures to different crystal faces may have an important effect on the gas-sensing properties of the samples.

In general, when the hydrothermal synthesis of tungsten oxide is assisted by soluble organic acids, the products usually form two-dimensional (2D) nanoplates or nanocubes [22,23]. In this experiment, we used the insoluble organic acid PABA as an auxiliary reagent in the hydrothermal synthesis of tungsten oxide, as shown in Figure 4. In the first step, the insoluble PABA reacts with sodium hydroxide to form a soluble salt, which is mixed with sodium tungstate to form a homogeneous solution. In the second step, after the introduction of hydrochloric acid, PABA and tungstic acid form a coprecipitation in the solution. In the third step, during the hydrothermal process, tungstic acid and PABA undergo a reversible dynamic equilibrium process of dissolution–crystallization. When the tungstic acid is recrystallized and dehydrated, it is easy to adsorb on the PABA crystal to form one-dimensional nanorods. In this process, PABA acts like a hard template. Na^+^ ions also act as an important capping agent. The strong electrostatic attraction between Na^+^ ions and negatively-charged tungsten species provide active sites for nanorod aggregation. Finally, when the solution system is cooled, both WO_3_·0.33H_2_O crystals and PABA are precipitated in the form of a eutectic solid. Because PABA is soluble in ethanol and tungstic acid is insoluble in ethanol, it is easy to obtain WO_3_·0.33H_2_O crystals by washing PABA with ethanol. Consequently, the WO_3_·0.33H_2_O nanorod bundles are built from nanorods to the second-order structure in a self-assembled manner.

Figure 5 shows the X-ray diffraction (XRD) analysis of the products. It can be seen that there is a significant difference between the products obtained by hydrothermal synthesis, with or without PABA. Figure 5 shows the XRD patterns (2θ = 10°–55°) of the hydrothermal product obtained without PABA. It can be seen from the figure that the XRD diffraction peak of S1 is the monoclinic WO_3_ (JCPDS Card No. 43-1035). The dominant peaks at 2θ = 23.11, 23.51, and 24.31 correspond to (002), (020), and (200) diffraction peaks of typical monoclinic WO_3,_ respectively. On the other hand, after the introduction of PABA, the products of the three samples after hydrothermal treatment are all WO_3_·0.33H_2_O (JCPDS Card No. 87-1203, space group: Fmm2 (No. 42)) crystal phase, but the intensity of some diffraction peaks is obviously different. The diffraction intensity ratios of (002) facets to (200) facets (i.e., I(002)/I(200)) of S1 and S3 are 1.26 and 3.32, respectively. Besides, the relative texture coefficient of a certain crystal facet (${\mathrm{TC}}_{\mathrm{hkl}}$) is defined to evaluate the degree of the crystal facet exposure. The texture coefficient of the (002) facets is given by Equation (4). Here, ${\mathrm{TC}}_{002}$ is the relative texture coefficient of the diffraction peaks of (002) over (200). $I_{\mathrm{hkl}}$ is the measured diffraction intensity of the (hkl) facet, and $I_{\mathrm{hkl}}^{0}$ is the corresponding value of the standard XRD pattern.
(4)$${\mathrm{TC}}_{002}=\frac{I_{002}/I_{002}^{0}}{I_{002}/I_{002}^{0}+I_{200}/I_{200}^{0}}$$

As calculated by Equation (4), the ${\mathrm{TC}}_{002}$ values of S1 and S3 are 0.622 and 0.296, respectively. Thus, the results discussed above indicate that S1 exposes a larger proportion of (002) facets and S3 exposes a larger proportion of (200) facets, consistent with the results of HRTEM.

### 3.2. Gas-Sensing Properties

In order to determine the optimum operating temperature for the determination of cyclohexene with four different amounts of PABA-synthesized tungsten oxide samples, the response (denoted here as response = *R*_g_/*R*_a_, in which *R*_a_ and *R*_g_ indicate the resistance of a sensor in air and in a tested gas, respectively) of a WO_3_-based sensor from 260 °C to 370 °C in 100 ppm gas was measured. As can be seen from Figure 6, the four samples have different responses at different temperatures for 100 ppm cyclohexene within 60 s. Among them, at 300 °C, several samples basically reached the maximum response value. S3 samples had the highest response value at 300 °C, at which point the response value reached 50.6. Therefore, we chose 300 °C as the optimal operating temperature.

In order to clarify the sensing characteristics of WO_3_·0.33H_2_O nanorod bundles, we further tested the influence of gas concentration on the gas-sensing properties of S3. Figure 7a shows the dynamic response and recovery curves of S3, corresponding to the response of 10–500 ppm cyclohexene at 300 °C. The response value varies from concentration changes and, thus, the curve exhibits a step-wise distribution. Due to the low concentration of gases, it takes a long time for the gas to reach the equilibrium state, when it diffuses to the surface active site. When the concentration of cyclohexene ranges from 10 ppm to 50 ppm, the gas response value of S3 to cyclohexene is between 5 and 12. When the concentration increases to 100 ppm, the response value increases sharply to over 50. When the concentration increases to 500 ppm, the response value can reach nearly 70. We also found that the response time of cyclohexene gas detection is 3 s. The response time is defined as the time taken for the relative conductance change to reach 90% of the steady-state value. The average recovery time is about 4 s. This indicates that the S3-based sensors have a rapid response and good reversibility to cyclohexene vapor.

The good selectivity to a target gas is an essential factor for gas sensors. The gas-sensing properties of the four as-synthesized samples were tested with a variety of inflammable or toxic gases, including 100 ppm toluene, styrene, o-xylene, methanol, cyclohexene, ethanol, acetone, ethyl acetate, and benzene at 300 °C (Figure 7b). As expected, the sensor based on WO_3_·0.33H_2_O nanorod bundles was found to exhibit a much higher responses to cyclohexene than the other gases. The result indicates that the WO_3_·0.33H_2_O nanorod bundles-based sensor apparently displays both higher response and better selectivity to cyclohexene gas.

Because there are few reports on the gas-sensing properties of cyclohexene, we compared the results from Table 1 with the main results reported at present. Compared with the previously reported cyclohexene gas sensor, the as-prepared WO_3_·0.33H_2_O sensor has the best gas-sensing performance. For example, under the conditions of 200 ppm cyclohexene and 240 °C, the gas sensitivity of CuO/ZnO nanocomposite was 2.1 [23], and that of CuO nanoplates was about 3 [24]. Under the conditions of 100 ppm cyclohexene and the optimum operating temperature of 370 °C, the gas sensitive response of WO_3_·0.33H_2_O nanoplates, synthesized by hydrothermal synthesis and assisted by p-nitrobenzoic acid, is only 8 [17]. It can be seen that WO_3_·0.33H_2_O nanorods prepared by this method have the best gas sensitivity to cyclohexene, compared with the gas-sensitive materials reported before. Although there is no conclusive evidence, we speculate that this is related to the exposure of different active crystal faces.

### 3.3. Discussion of Gas-Sensing Mechanism

The enhanced gas-sensing performance of the WO_3_·0.33H_2_O nanorod bundles may be attributed to a large number of active sites and space for adsorption, as well as the reaction between target gases and adsorbed oxygen ions [25]. When the WO_3_·0.33H_2_O-based sensor was exposed to air, oxygen molecules could be absorbed on the nanorod bundles to form ions by capturing electrons from the conductance band, which reduced concentration of the free carrier and formed an electron depletion layer. The species of the adsorbed ionic oxygen is strongly reliant on the reaction temperature, we assume that adsorbed ionic oxygen was not transformed under the optimal working temperature of 300 °C, and the detailed reaction processes are listed as follows:
(5)$$O_{2\left(\mathrm{gas}\right)}\;↔{\;O}_{2\left(ads\right)}$$
(6)$$O_{2\left(\mathrm{ads}\right)}\;+{\;e}^{-}↔{\;O}_{2\left(ads\right)}{}^{-}$$
(7)$$O_{2\left(ads\right)}{}^{-}\;+{\;e}^{-}\;↔\;2{\;O}_{\left(ads\right)}{}^{-}$$

When the gas sensor is exposed to reducing gases such as cyclohexene, the oxygen ions adsorbed on the surface of nanorods will react with cyclohexene. The adsorbed electrons are then released into the WO_3_·0.33H_2_O nanorod bundles, resulting in a decrease in resistance and an increase in gas sensitivity. We proposed a possible reaction of cyclohexene with the adsorbed ions, which is listed as follows:(8)$$C_{6}H_{10\left(\mathrm{ads}\right)}\;+\;17{\;O}_{\left(ads\right)}{}^{-}\;\to \;6{\;\mathrm{CO}}_{2\left(g\right)}\;+\;5{\;H}_{2}O_{\left(g\right)}\;+\;17{\;e}^{-}$$

On the other hand, as discussed in the TEM and XRD section, the WO_3_·0.33H_2_O nanorod bundles feature exposed (200) facets, which lead to high sensitivity and selectivity compared to other exposed facets. There are more oxygen vacancies in the (200) polar plane of WO_3_·0.33H_2_O than other facets. These oxygen vacancies can significantly enhance the adsorption of oxygen molecules and promote dissociative adsorption, in which one of the oxygen atoms fills the original oxygen vacancy while the other forms a bridging oxygen atom between two W atoms. Therefore, C_6_H_10_ molecules are easier to adsorb and react on the (200) facets to form more oxygen vacancies, resulting in faster charge transfer between C_6_H_10_ molecules and our sensing materials and enhancing the properties of detection C_6_H_10_ at low operating temperatures. We achieved a fast and selective response by using our WO_3_·0.33H_2_O nanorod bundles sensor. Therefore, it was found that the gas-sensing properties deeply depend on the exposed facet and grain size, and the dominant facet effect masks the size effect for nanocrystals.

## 4. Conclusions

In this paper, WO_3_·0.33H_2_O nanorods were synthesized by a hydrothermal method with PABA as an organic mineralizer. When no mineralizer or auxiliary reagent was used, only monoclinic WO_3_ nanoparticles were obtained after the hydrothermal treatment of tungstic acid precursors. However, when different amounts of PABA were introduced into the hydrothermal system, orthogonal WO_3_·0.33H_2_O nanorods were obtained. Moreover, HRTEM results showed that the four products exposed different crystal faces, which had an important influence on the gas-sensing properties. The results showed that the WO_3_·0.33H_2_O nanorods synthesized with 8 g of PABA had the highest gas sensitivity. At 300 °C, the gas sensitivity of the WO_3_·0.33H_2_O nanorods to 100 ppm cyclohexene could reach as high as 50, which is better than that reported in the literature. The results indicate that the WO_3_·0.33H_2_O nanorod bundles may have great application prospects in commercial and industrial real-time cyclohexene-detecting or -monitoring systems.

## Figures and Tables

**Figure 1 sensors-19-01257-f001:**
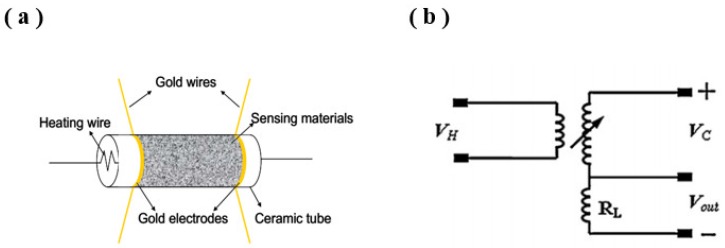
(**a**) The structural sketch of the WO_3_·0.33H_2_O-based gas sensor and (**b**) the measuring electric circuit of the sensor.

**Figure 2 sensors-19-01257-f002:**
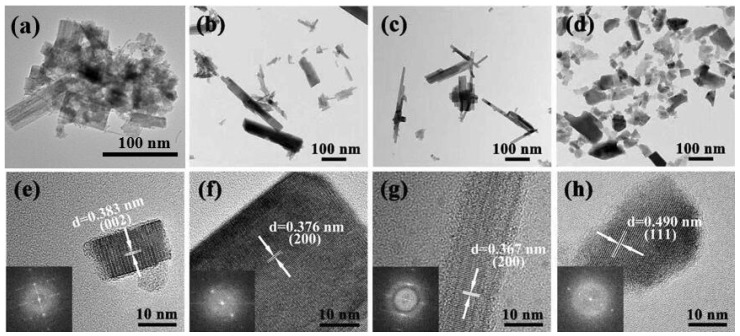
(**a**–**d**) TEM images of S1, S2, S3, S4; (**e**–**h**) HRTEM images of S1, S2, S3, S4.

**Figure 3 sensors-19-01257-f003:**
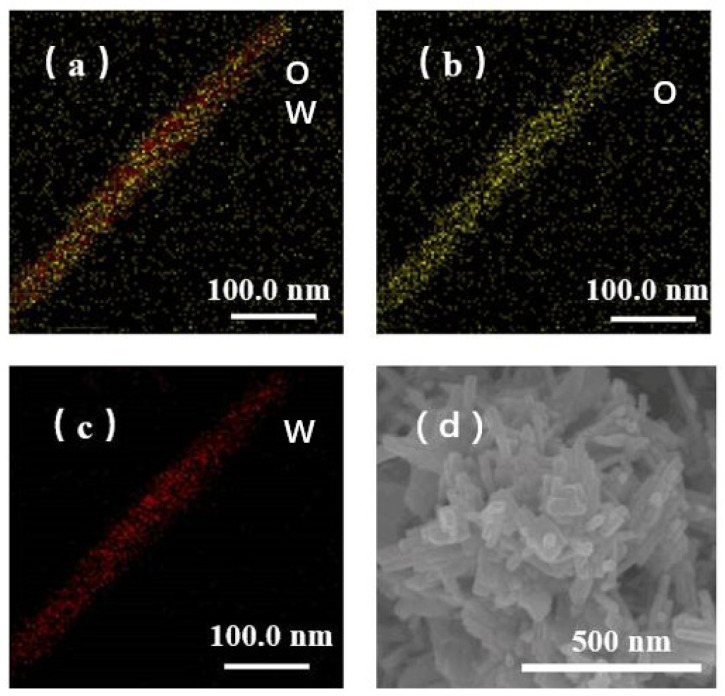
(**a**–**c**) Elemental mapping of S3; (**d**) FESEM image of S3.

**Figure 4 sensors-19-01257-f004:**
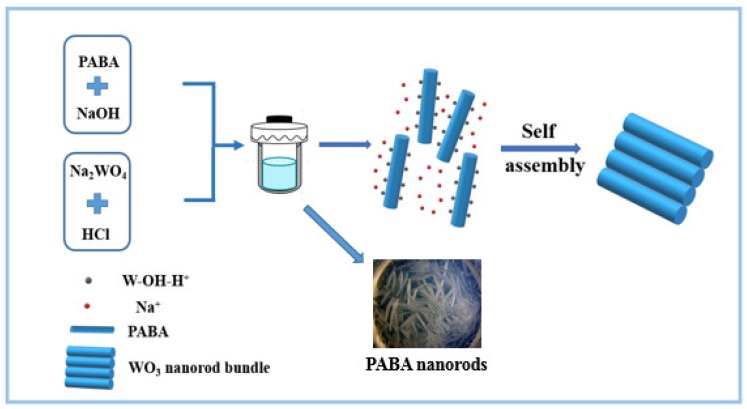
Schematic illustration of WO_3_·0.33H_2_O nanorod bundles through a self-assembly step.

**Figure 5 sensors-19-01257-f005:**
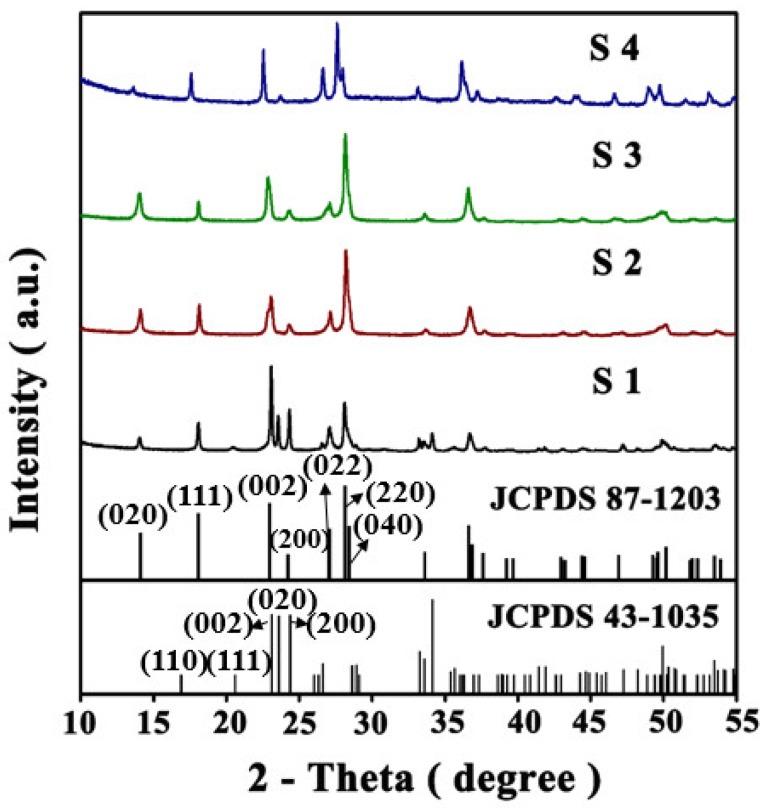
X-ray diffraction (XRD) patterns (2θ = 10°–55°) of S1, S2, S3, S4.

**Figure 6 sensors-19-01257-f006:**
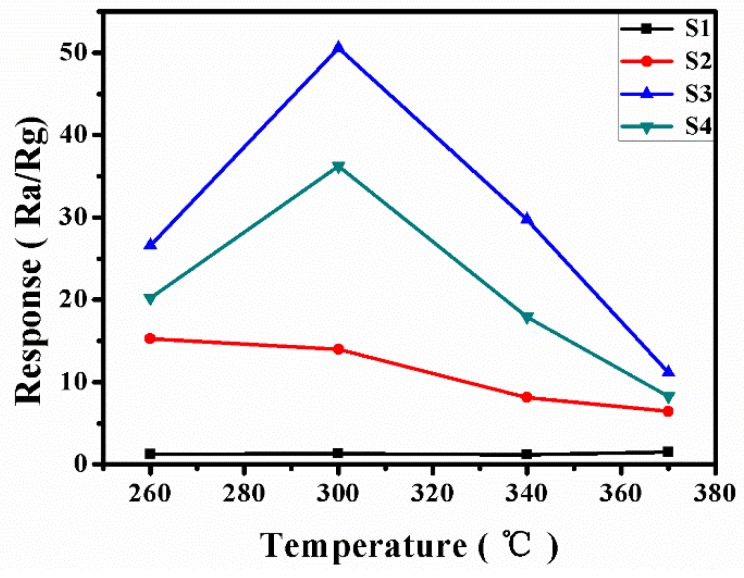
The responses of the samples (S1–S4) to 100 ppm cyclohexene vapor at different temperatures.

**Figure 7 sensors-19-01257-f007:**
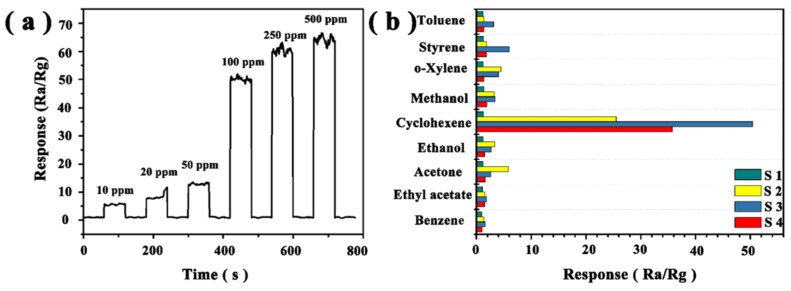
(**a**) Gas response recovery of S3 toward cyclohexene within the concentration range of 10–500 ppm at 300 °C; (**b**) gas response of WO_3_·0.33H_2_O-based samples toward 100 ppm of different test gases at 300 °C.

**Table 1 sensors-19-01257-t001:** List of metal oxide-based ethanol sensors.

Material	Concentration (ppm)	Response	T (°C)	Reference
CuO/ZnO nanocomposite	200	2.1	240	[15]
CuO nanoplates	200	3	240	[16]
WO_3_·0.33H_2_O nanoplates	100	8	370	[9]
WO_3_ nanoplates	1000	140	250	[7]
WO_3_·0.33H_2_O nanorod bundles	100	50.6	300	our work

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
