# Peer review of "Hydrothermal Synthesis of WO3·0.33H2O Nanorod Bundles as a Highly Sensitive Cyclohexene Sensor"

_sensors, 2019, doi:10.3390/s19051257_

Round 1
Reviewer 1 Report
The article “Hydrothermal synthesis of WO3·0.33H2O nanorod bundles as highly sensitive cyclohexene sensor” is an interesting, well-written paper reporting a novel variant of the hydrothermal synthesis of WO3·0.33H2O and interestingly high sensing abilities of the synthesized material. The gas sensing experiments are carried out well, and the sensing capabilities of the materials are promising. The manuscript should be published after major revisions. All points requiring revision are summarized in the following. The biggest bottleneck is the used additive PABA and the authors comments of some aspects of PABA (I’ll comment in detail with the last point of the following list). The following points need to be addressed by the authors:
- Abstract: Why mention the amount on how much of the additive p-aminobenzoic acid (PABA) was used? I find it bizarre that the authors give “8g” in brackets (line 16). This is quite a specific information for being in the abstract. Plus “8g” is not a helpful measure = the amount would have a different effect in a hydrothermal (HT) synthesis carried out at different volume. If this musty be mentioned, concentration would make more sense.
- In the introduction, the authors nicely motivate why WO3*.33H2O is an interesting gas sensing material for cyclohexene, which in turn is a compound for which sensor are required. However, there are already HT syntheses for WO3*.33H2O reported (using other additives). The authors motivate their study by: “…would be the interesting results if some poorly soluble organic acids, such as p-aminobenzoic acid (PABA)”. But why would this be interesting? Just because nobody ever did it? What outcomes did the authors expect by using a “poorly soluble acid”? Please motivate the choice of synthetic approach
- Experimental part needs improvement: For an HT experiment the filling ratio of the autoclave is very important (for the liquid H2O to H2O vapor ratio). Therefore: please give the volume or mass of sodium p-aminobenzoate solution. Also, give volume or mass of added HCl necessary to adjust pH to 1. Why is the pH adjusted to 1? Please comment.
- Methods: XRD description. 2o min-1 should read 2 {degree sign=^o}(2 theta) min-1. It’s important to give 2theta in brakets after the degree sign. The used X-ray source is presumably not Cu-K but Cu-K_alpha (at least fron the given wavelength)
- Structural characterization: Fig 2a needs a zoom into some of the particles. The form is not discernable from the displayed TEM image.
- TEM: the authors write that the nanorods have a “strong surface interaction capability” (line 109). Can they please clarify? Nanoparticles have a high surface atom to bulk atom ratio. The surfaces is NPs are rich in “dangling bonds” and therefore NPs have a tendency to stack together (this is also visible in Fig. 2a for the non-rodlike shapes). The stacking the authors observe are not surprising (especially given their anisotropic form, which favors the type of observed packing). I do not understand why the observed stacking/packing should be special.
- Mechanistic scheme (Fig. 3). The authors need to label this scheme clearly as a hypothesis, since according to what is presented the “mechanism” is not based on a mechanistic study, and it is not proven. It’s a hypothesis and it needs to be labeled as such. What is also unclear from the manuscript is on what the authors base their mechanistic hypothesis. Please explain why you think the proposed events happen in your reaction. Moreover, the mechanistic scheme lacks the chloride ions that were added through adjusting the pH with HCl. Where is the photograph of the “PABA nanorods” in the mechanistic picture coming from? Is this the aspect of the product mixture after the HT synthesis?
- TEM description. One of the given lattice spacings 0490 should read 0.490 nm
- Sensor preparation using EtOH as binder. It would be helpful to have an SEM image of what a sensor looks like. Are the nanorods tightly packed after the preparation of the sensor? Are they overlapping or rather tiling next to each other on the alumina surface? This information would also be very helpful for hypothising on the sensor performance. At sucha good performance, I would expect that the majority of particle facets are accessible (not too many particles stacking on top of each other)
- comparison of cyclohexene sensors. I suggest that the authors add a comparison table that gives sensing performance, used sensor material, sensing temperature and conditions. That would be easier to read than the current comparison in plane text only.
- Finally and most importantly: the additive PABA. The authors need to comment much more on PABA and revise their comments on PABA. PABA is not just an organic acid, it also bears an amino group. Hence, one would expect the formation of an internal salt p-ammonium benzene carboxylate in H2O. This can be easily verified through the pKAs of the NH2 and the CO2H function. Eventually, this type of internal salt is not forming as pH is adjusted to 1 with HCl. In any case, the authors need to comment on the solubility. Moreover, no matter if in the internal salt form or not, I would expect PABA to be well soluble in water under hydrothermal conditions (and not to be a poorly soluble acid as the authors write). There is a bulk of work where aromatic amines and acids are used as actual starting compounds in hydrothermal reactions and in these reports they are well soluble hydrothermally. See e.g. (1) Baumgartner et al 2014 10.1039/C4PY00263F ; (2) Baumgartner et al 2015 10.1039/C5PY00231A (3) Baumgartner 2016 https://doi.org/10.1002/macp.201500287 {this paper reports internal salts of the ammonium carboxylate types}; The compounds that are soluble in HT conditions can be surprisingly apolar even rylene derivatives: (4) Baumgartner et al 2017 10.1039/C6CC06567H; and (5) Taublaender et al 2018 10.1002/anie.201801277. Organic compounds that have little solubility in water become soluble in HT through the properties of water at higher T: (6) Unterlass 2017 https://doi.org/10.3390/biomimetics2020008
Author Response
Sensors—438280
Title: Hydrothermal synthesis of WO3·0.33H2O nanorod bundles as highlysensitive cyclohexene sensorAuthors: Xiaofei Liu, Xintai Su, Chao Yang, Kongjun Ma *
Dear Editor and Reviewers:
Thank you for kindly considering out our manuscript entitled of “Hydrothermal synthesis of WO3·0.33H2O nanorod bundles as highly sensitive cyclohexene sensor” for publishing in Sensors. We appreciate very much the positive comments from the two referees. Their comments are very helpful for us to improve the quality of the paper.According to the comments of the referees, we have carefully revised the manuscript. All changes were highlighted in yellow in the revised manuscript. We hope that we have successfully addressed the referees’ comments in a way that will ensure the publication of this paper.
We sincerely thank you for your time and effort in handling this paper. Additionally, it will be great if you could inform us at your early convenience when it is received or accepted.
Sincerely yours,
Kongjun Ma
Response to Comments
Reviewer #1:
The article “Hydrothermal synthesis of WO3·0.33H2O nanorod bundles as highly sensitive cyclohexene sensor” is an interesting, well-written paper reporting a novel variant of the hydrothermal synthesis of WO3·0.33H2O and interestingly high sensing abilities of the synthesized material. The gas sensing experiments are carried out well, and the sensing capabilities of the materials are promising. The manuscript should be published after major revisions. All points requiring revision are summarized in the following. The biggest bottleneck is the used additive PABA and the authors comments of some aspects of PABA (I’ll comment in detail with the last point of the following list). The following points need to be addressed by the authors:
- Abstract: Why mention the amount on how much of the additive p-aminobenzoic acid (PABA) was used? I find it bizarre that the authors give “8g” in brackets (line 16). This is quite a specific information for being in the abstract. Plus “8g” is not a helpful measure = the amount would have a different effect in a hydrothermal (HT) synthesis carried out at different volume. If this musty be mentioned, concentration would make more sense.
RE: Thank you very much for your valuable advice. Because the 8 g PABA sample has better gas sensitivity, we have marked it in brackets. As PABA is a insoluble solid, it is difficult to express it in terms of concentration. However, it is true that the use of such a specific number in the summary is not necessary, so, in your opinion, we have removed the 8 g in brackets.
- In the introduction, the authors nicely motivate why WO3·0.33H2O is an interesting gas sensing material for cyclohexene, which in turn is a compound for which sensor are required. However, there are already HT syntheses for WO3·0.33H2O reported (using other additives). The authors motivate their study by: “…would be the interesting results if some poorly soluble organic acids, such as p-aminobenzoic acid (PABA)”. But why would this be interesting? Just because nobody ever did it? What outcomes did the authors expect by using a “poorly soluble acid”? Please motivate the choice of synthetic approach.
RE: We greatly appreciate your kindly comments. Soluble organic acids, such as citric acid, tartaric acid, ascorbic acid, have been used in hydrothermal synthesis of tungsten oxide. They have important effects on the morphology and phase of tungsten oxide products. However, the solubility of poorly soluble organic acids is lower at room temperature, and the solubility of poorly soluble organic acids can be greatly increased under high temperature and high pressure. Our experimental results also show that the product is WO3·0.33H2O, and the exposed active crystal faces are different when different amounts of PABA are used.
- Experimental part needs improvement: For an HT experiment the filling ratio of the autoclave is very important (for the liquid H2O to H2O vapor ratio). Therefore: please give the volume or mass of sodium p-aminobenzoate solution. Also, give volume or mass of added HCl necessary to adjust pH to 1. Why is the pH adjusted to 1? Please comment.
RE: Thank you for your valuable advice. It is true that the degree of filling has an important impact on hydrothermal synthesis. In the experiment, the volume of the solution put into the reactor is 60 mL. In the article, we also made the necessary explanation and modification.At pH 1, the acidity of tungstic acid is sufficient to precipitate. Therefore, many literatures use hydrochloric acid to adjust the acidity when hydrothermal synthesis of tungsten oxide.
- Methods: XRD description. 2o min-1 should read 2 {degree sign=^o}(2 theta) min-1. It’s important to give 2theta in brakets after the degree sign. The used X-ray source is presumably not Cu-K but Cu-K_alpha (at least fron the given wavelength)
RE: As reviewer suggested, the error have been corrected in line 67.
- Structural characterization: Fig 2a needs a zoom into some of the particles. The form is not discernable from the displayed TEM image.
RE: Thanks for your comments. We have changed a clear picture of magnification, as shown in Fig. 2a.
- TEM: the authors write that the nanorods have a “strong surface interaction capability” (line 109). Can they please clarify? Nanoparticles have a high surface atom to bulk atom ratio. The surfaces is NPs are rich in “dangling bonds” and therefore NPs have a tendency to stack together (this is also visible in Fig. 2a for the non-rodlike shapes). The stacking the authors observe are not surprising (especially given their anisotropic form, which favors the type of observed packing). I do not understand why the observed stacking/packing should be special.
RE: Thanks to your suggestion, due to the large surface area of nanoparticles, stacking/packing is indeed a relatively common phenomenon. Therefore, based on your suggestion, we have modified the corresponding description in the article.
- Mechanistic scheme (Fig. 3). The authors need to label this scheme clearly as a hypothesis, since according to what is presented the “mechanism” is not based on a mechanistic study, and it is not proven. It’s a hypothesis and it needs to be labeled as such. What is also unclear from the manuscript is on what the authors base their mechanistic hypothesis. Please explain why you think the proposed events happen in your reaction. Moreover, the mechanistic scheme lacks the chloride ions that were added through adjusting the pH with HCl. Where is the photograph of the “PABA nanorods” in the mechanistic picture coming from? Is this the aspect of the product mixture after the HT synthesis?
RE: Chloride ions do have some effect on hydrothermal synthesis of tungsten oxide, but acidity has a greater effect. The photo of PABA nanorods in the mechanical drawing is the PABA crystal which is crystallized on the surface of the solution after hydrothermal treatment. The size of the PABA crystal is relatively large and can be seen by naked eyes. This photo was taken by a digital camera. After hydrothermal treatment, PABA crystals suspended in solution can be removed by filtration and ethanol cleaning.- TEM description. One of the given lattice spacings 0490 should read 0.490 nm RE: Thank you for your amendment, the error have been corrected.
- Sensor preparation using EtOH as binder. It would be helpful to have an SEM image of what a sensor looks like. Are the nanorods tightly packed after the preparation of the sensor? Are they overlapping or rather tiling next to each other on the alumina surface? This information would also be very helpful for hypothising on the sensor performance. At such a good performance, I would expect that the majority of particle facets are accessible (not too many particles stacking on top of each other)
RE: We have supplemented the SEM, and have done the corresponding analysis and discussion in the article.
- comparison of cyclohexene sensors. I suggest that the authors add a comparison table that gives sensing performance, used sensor material, sensing temperature and conditions. That would be easier to read than the current comparison in plane text only.
RE: At line 200, We added a table of the performance of cyclohexene gas sensors.
- Finally and most importantly: the additive PABA. The authors need to comment much more on PABA and revise their comments on PABA. PABA is not just an organic acid, it also bears an amino group. Hence, one would expect the formation of an internal salt p-ammonium benzene carboxylate in H2O. This can be easily verified through the pKAs of the NH2 and the CO2H function. Eventually, this type of internal salt is not forming as pH is adjusted to 1 with HCl. In any case, the authors need to comment on the solubility. Moreover, no matter if in the internal salt form or not, I would expect PABA to be well soluble in water under hydrothermal conditions (and not to be a poorly soluble acid as the authors write). There is a bulk of work where aromatic amines and acids are used as actual starting compounds in hydrothermal reactions and in these reports they are well soluble hydrothermally. See e.g. (1) Baumgartner et al 2014 10.1039/C4PY00263F ; (2) Baumgartner et al 2015 10.1039/C5PY00231A (3) Baumgartner 2016 https://doi.org/10.1002/macp.201500287 {this paper reports internal salts of the ammonium carboxylate types}; The compounds that are soluble in HT conditions can be surprisingly apolar even rylene derivatives: (4) Baumgartner et al 2017 10.1039/C6CC06567H; and (5) Taublaender et al 2018 10.1002/anie.201801277. Organic compounds that have little solubility in water become soluble in HT through the properties of water at higher T: (6) Unterlass 2017 https://doi.org/10.3390/biomimetics20200081. It is better to analyze the elemental composition of the two kinds of wastewater in the manuscript.
RE: Thank you for your question. In this experiment, we used the insoluble organic acid-PABA as auxiliary reagent in hydrothermal synthesis. The insoluble PABA reacts with sodium hydroxide to form a soluble salt, which is mixed with sodium tungstate to form a homogeneous solution. And in the hydrothermal process, tungstic acid and PABA undergo a reversible dynamic equilibrium process of dissolution-crystallization. Thank you for your suggestion, the literatures have been cited, these literatures have a great effect on the persuasiveness of my manuscript.
Reviewer 2 Report
In the manuscript SENSORS-438240, the authors have prepared WO3·0.33H2O nanorods by hydrothermal method with PABA as organic mineralizer and demonstrated the detection of cyclohexene. My decision is ‘major revision’ and the authors need to address the following concerns:
1. The XRD peaks should be indexed in Fig 4.
2. Line 97, Caption of Fig 2. Where is the HRTEM image of S4 or do the authors forget to write here.
3. How do the authors confirm that the prepared sample contains 0.33 H2O. XPS analysis could be beneficial to illustrate this.
4. Elemental mapping should also be provided to determine the purity of the samples.
5. Authors are using various terms (sensitivity, response) at various places. The denotation term should be the same. Eg Line 163 and 164.
6. What is the reason for higher selectivity to cyclohexene in Fig 6b.
7. What is the value for response/recovery time from Fig 6a
8. The mechanism is very poorly discussed. What is the benefit of rod like morphology and what is the role of PABA in enhanced sensing towards cyclohexene. These aspects should also be evaluated.9. Also it is not clear why the sample prepared by using 8g PABA shows highest response.
10. Introduction section is poorly written. The authors are advised to increase the number of references while properly highlighting the novelty of this work in the introduction section. The authors should explain their point of view to carry-out this work and the utility of this technique than earlier reported ones.
11. It is advised to include the following relevant references: 10.1039/C8TA02702A; 10.1016/j.snb.2017.06.179, 10.1016/j.snb.2016.08.022
Author Response
Sensors-438280
Title: Hydrothermal synthesis of WO3·0.33H2O nanorod bundles as highlysensitive cyclohexene sensorAuthors: Xiaofei Liu, Xintai Su, Chao Yang, Kongjun Ma *
Dear Editor and Reviewers:
Thank you for kindly considering out our manuscript entitled of “Hydrothermal synthesis of WO3·0.33H2O nanorod bundles as highly sensitive cyclohexene sensor” for publishing in Sensors. We appreciate very much the positive comments from the two referees. Their comments are very helpful for us to improve the quality of the paper.According to the comments of the referees, we have carefully revised the manuscript. All changes were highlighted in yellow in the revised manuscript. We hope that we have successfully addressed the referees’ comments in a way that will ensure the publication of this paper.
We sincerely thank you for your time and effort in handling this paper. Additionally, it will be great if you could inform us at your early convenience when it is received or accepted.
Sincerely yours,
Kongjun Ma
Response to Comments
Reviewer #2:
In the manuscript SENSORS-438240, the authors have prepared WO3·0.33H2O nanorods by hydrothermal method with PABA as organic mineralizer anddemonstrated the detection of cyclohexene. My decision is ‘major revision’ and the authors need to address the following concerns:
1. The XRD peaks should be indexed in Fig 4.
RE:As Reviewer suggested,the XRD peaks have been indexed in Fig. 4.
2. Line 97, Caption of Fig 2. Where is the HRTEM image of S4 or do the authors forget to write here.
RE:We are very sorry for our negligence, this problem have been corrected (Page 3, line 98).
3. How do the authors confirm that the prepared sample contains WO3·0.33H2O XPS analysis could be beneficial to illustrate this.
RE: Thank you for your advice, the elemental mapping can also confirm that the prepared sample contains WO3·0.33H2O .
4. Elemental mapping should also be provided to determine the purity of the samples.
RE: We have supplemented the elemental mapping, and have done the corresponding analysis and discussion in the article.
5. Authors are using various terms (sensitivity, response) at various places. The denotation term should be the same. Eg Line 163 and 164.
RE: We greatly appreciate your proposed suggestion, the problem has been corrected in line 175.
6. What is the reason for higher selectivity to cyclohexene in Fig 6b.
RE: The higher selectivity to cyclohexene maybe relate to the (200) facets which WO3·0.33H2O nanorod bundles exposed.
7. What is the value for response/recovery time from Fig 6a.
RE: Thank you for this question, response time and recovery time have been mentioned in line 178-181.
8. The mechanism is very poorly discussed. What is the benefit of rod like morphology and what is the role of PABA in enhanced sensing towards cyclohexene. These aspects should also be evaluated.
RE: PABA just as structural agents for hydrothermal synthesis of the WO3·0.33H2O nanorod bundles. The WO3·0.33H2O nanorod bundles may be attributed to a large amount of active sites and space for adsorption and reaction between target gases and adsorbed oxygen ions. Based on the experimental results, we consider that PABA affects the crystal plane of the product, and the crystal plane may be the most important factor affecting the gas sensing properties.
9. Also it is not clear why the sample prepared by using 8g PABA shows highest response.
RE: When 8g PABA is added, the WO3·0.33H2O nanorod bundles are formed and exposed the (200) facets, the gas-sensing properties depend deeply on the exposed facet and grain morphology.
10. Introduction section is poorly written. The authors are advised to increase the number of references while properly highlighting the novelty of this work in the introduction section. The authors should explain their point of view to carry-out this work and the utility of this technique than earlier reported ones.
RE:Thank the reviewer for the comments. We have carefully revised the preface and added some important references.
11. It is advised to include the following relevant references:10.1039/C8TA02702A; 10.1016/j.snb.2017.06.179, 10.1016/j.snb.2016.08.022
RE: Thank you for your suggestion, the literature has been cited.
Round 2
Reviewer 1 Report
The manuscript has improved much by the revisions performed by the authors. It is now a clearer read, which makes me believe that the authors' results will be more appreciable by the community. I found two minor typos/text issues. As I think they will be addressed in the final type setting of the manuscript by the editorial office, I do not request minor revisions but recommend "acceptance as is". These are the two points:
(1) page 2, line 67: Cu-K should read Cu-Kα
(2) reference section, page 9: references are inconsistently given. Some references give full author names (e.g. ref. 5), others author name initials only (also ref 5, after 1st author only initials are given), others give first names and initial of last name (e.g. ref. 8). Please check.
Finally, I wish to congratulate the authors for their work and hope for a positive editorial decision.
Reviewer 2 Report
The authors have answered to all my queries, hence, my decision is manuscript accept.